civil engineering

climate change, regional climate model, representative concentration pathways, SWAT

**Authors for correspondence:**
Muhammad Izhar Shah
e-mail: mizhar@cuiatd.edu.pk
Tahir Ali Akbar
e-mail: drtahir@cuiatd.edu.pk

# Predicting hydrologic responses to climate changes in highly glacierized and mountainous region Upper Indus Basin

Muhammad Izhar Shah[1,2], Asif Khan[1], Tahir Ali Akbar[2], Quazi K. Hassan[4], Asim Jahangir Khan[3] and Ashraf Dewan[5]

[1]Department of Civil Engineering, University of Engineering and Technology (UET), Peshawar, Pakistan
[2]Department of Civil Engineering, and [3]Department of Environmental Sciences, COMSATS University Islamabad, Abbottabad Campus, Abbottabad 22060, Khyber Pakhtunkhwa, Pakistan
[4]Department of Geomatics Engineering, Schulich School of Engineering, University of Calgary, Calgary, Alberta, Canada
[5]Spatial Sciences Discipline, School of Earth and Planetary Sciences, Curtin University, Kent St, Bentley, WA 6102, Australia

MIS, 0000-0002-0588-6301; AK, 0000-0002-5731-9157;
TAA, 0000-0001-5385-4120

The Upper Indus Basin (UIB) is a major source of supplying water to different areas because of snow and glaciers melt and is also enduring the regional impacts of global climate change. The expected changes in temperature, precipitation and snowmelt could be reasons for further escalation of the problem. Therefore, estimation of hydrological processes is critical for UIB. The objectives of this paper were to estimate the impacts of climate change on water resources and future projection for surface water under different climatic scenarios using soil and water assessment tool (SWAT). The methodology includes: (i) development of SWAT model using land cover, soil and meteorological data; (ii) calibration of the model using daily flow data from 1978 to 1993; (iii) model validation for the time 1994–2003; (iv) bias correction of regional climate model (RCM), and (v) utilization of bias-corrected RCM for future assessment under representative concentration pathways RCP4.5 and RCP8.5 for mid (2041–2070) and late century (2071–2100). The results of the study revealed a strong correlation between simulated and observed flow with $R^2$ and Nash–Sutcliff efficiency (NSE)

equal to 0.85 each for daily flow. For validation, $R^2$ and NSE were found to be 0.84 and 0.80, respectively. Compared to baseline period (1976–2005), the result of RCM showed an increase in temperature ranging from 2.36°C to 3.50°C and 2.92°C to 5.23°C for RCP4.5 and RCP8.5 respectively, till the end of the twenty-first century. Likewise, the increase in annual average precipitation is 2.4% to 2.5% and 6.0% to 4.6% (mid to late century) under RCP4.5 and RCP8.5, respectively. The model simulation results for RCP4.5 showed increase in flow by 19.24% and 16.78% for mid and late century, respectively. For RCP8.5, the increase in flow is 20.13% and 15.86% during mid and late century, respectively. The model was more sensitive towards available moisture and snowmelt parameters. Thus, SWAT model could be used as effective tool for climate change valuation and for sustainable management of water resources in future.

# 1. Introduction

The water resources availability is associated with the comfort and well-being of human societies that need it for drinking, agriculture and industrial activities. The water resources are largely affected by precipitation, depletion of aquifer and droughts. More than 50% of world water demand is fulfilled by the rivers [1] but these are more vulnerable because of variations in temperature and precipitation specifically in snowmelt areas [2,3]. Moreover, the climate system of the Earth was changed up to large extent in the past [4]. The snow/glacial periods alternated with (warmer) inter-glacial periods. On a global scale, warming of the atmospheric system was observed since the last century. In the past century, an increase occurred in ocean and overall mean air temperature causing large quantity of ice/snowmelt and consequently, overall average sea level rose [5,6]. On average, an increase of 0.74°C on Earth's surface temperature was observed during the past 100 years (1906–2005) and therefore, global warming had become an undeniable fact [4]. The global hydrological cycle is changing constantly because of global warming. The increased water vapour content in the atmosphere, changes in precipitation patterns, alteration in snowmelt-fed rivers and warming of rivers and lakes might increase evaporation [5].

Pakistan is an agrarian country and to a large extent, and its economy depends on agriculture. The country is reliant on irrigation by spreading an immense system of channels, barrages and diversion structures fed by the River Indus and its tributaries. The accessibility and intensity for hydroelectricity and irrigation is likely to be affected by climate changes [7]. Melted water from glaciers in Karakoram, Himalayas and Hindu-Kush highlands and rainfall-induced runoff [8] during winter and summer months covering 2200 km$^2$ of the permanently glaciated area, ultimately fed the Indus River system [9]. As far as the water resources and reservoirs' lives are concerned, the knowledge and interaction of the hydrological systems of the mountains are therefore of utmost importance for the country like Pakistan [10]. The country is currently undergoing the regional impacts of global climate change. During the last few years, severe storms and floods have occurred in its history. The rapid population growth and the accompanying land-use alterations further intensified the problems induced by climate change. The surface temperature also increases due to land-use changes [11]. Therefore, preparation for such extreme events requires investigation and further research in hydrological modelling and climatic change studies. The only possibility is to well predict the future climate and hydrological conditions earlier in order to make the necessary adaptations and reduce the potential damage [10].

The impacts of climate changes on water resources vary from basin to basin because of the complexity of hydro-climatic regimes [12]. The potential effects of climate changes on hydrological processes are variations in water temperature, evapotranspiration, stream-flow volume, soil moisture, frequency and magnitude of runoff and frequency of floods. These changes in hydrological regimes would ultimately affect different important aspects such as water supply, agricultural productivity, power generation and the biotic ecosystem [13,14]. The climate change studies and global hydrological cycles play a significant role as the results from these studies are helpful in better understanding and planning effective strategies for sustainable management of water resources. The different studies were conducted to estimate the possible climate changes, snow/glaciers melt and related hydrological interaction in the Upper Indus Basin (UIB) and surrounding catchments. Lutz et al. [3] used high-resolution cryospheric hydrological model to estimate the upstream hydrological regimes in UIB. The results revealed that stream flow was mainly due to glacier meltwater, contributing approximately 40% of the total runoff which exposed variation in future hydrology. Babur et al. [15] considered the climate change impacts on Mangla reservoir discharge using a set of global circulation models (GCMs) and simulate soil and water assessment tool (SWAT) model. The model was used to simulate stream flow for early, mid, and late century under representative concentration

pathways RCP4.5 and RCP8.5 scenarios. The results indicated increase in mean annual stream flow and the estimated high flows tends to increase along with decrease in middle flows. The spring and summer season also showed a considerable increase in stream flow. The shift in peak flows were also observed from May to July. Conclusively, the reservoir could face more flood in future because of probable increase in stream flow, and a great variation in peak flow. The climate change effect on water resources due to snowmelt was studied by Tahir *et al.* [16] using the snowmelt-runoff model (SRM) along with MODIS remote sensing snow-cover data in Hunza River Basin. The results of the SRM under future climate showed almost doubling the summer runoff till the mid-century. Shrestha *et al.* [8] revealed that runoff in the Hunza river was strongly influenced by the snow and glacier melt with almost 50%, 33% and 17% contribution from snow, glacier melt and rainfall respectively. Garee *et al.* [14] used SWAT model with five general circulation models (GCMs) in Hunza basin. The study revealed an increase in temperature and precipitation up to 6.5°C and 31%, respectively, with increased variation in surface runoff ranging from 5% to 10%. Anjum *et al.* [17] studied the implications of changing climate in Swat River basin using SWAT along with six GCM models under two RCPs (4.5 and 8.5). The outcomes of the study showed temperature increase up to 4.18°C and 8.49°C and precipitation rise by 22.52% and 35.98% for RCP4.5 and 8.5, respectively. Moreover, increase in annual average flow ranging from 0.3% to 44.4% under both RCPs. An increase in middle and low flows were also observed with a shift in peak flow from July to June. Hence, it was suggested that consideration of climate change-induced effects is indispensable for effective planning and management of hydropower projects.

The precipitation in the Himalaya region remains poorly defined due to lack of reliable rainfall networks [18] and it does not provide a true depiction of the area particularly for elevation zones. Data for longer duration is only available at some stations because functioning of these stations starts after the mid-1990s. Similarly, the precipitation system of the region cannot be fully explained by thin observed station data or the sensor-based data because of the complex orography and the interaction of different hydro-climatic regimes [19–21]. Most hydrological investigations relied only on limited observation stations data because of the limitation of long-term data. In such situations, data have shorter duration but are of acceptable quality, or may have large variations, specifically with higher altitudes precipitation regions. The calculations of water balance and spatially distributed rainfall-runoff models require high-resolution climatic datasets, i.e. temperature and precipitation. The UIB is facing the same problem because of valley-based gauging stations, which cause temporal discontinuities and are unable to capture orographic influences. Therefore, the climate studies in such mountainous regions are facing the water imbalances [22].

However, considering the above limitation and non-availability of reliable data to portray the processes and truly describe the study area, the present study used the fully distributed, regionalized and corrected precipitation and temperature data [22], which were interpolated to sub-basin centroids and constructed based on true situation in UIB. The datasets were corrected by incorporating the orographic effect and improving the influences induced by higher elevation, available runoff data, glacier mass-balance and actual evapotranspiration. The precipitation data are the most important inputs in modelling studies of mountainous regions [18,23], and the results are strongly affected because of uncertainty in spatial distribution [22]. Furthermore, in this study, the elevation bands were used for better representation of the melt processes and assigned distributed melt parameters to different bands and sub-basins.

# 2. Study area and data collection

## 2.1. Study area

The Indus River can be classified as one of the major rivers in Asia, covering approximately 2880 km length and draining 912 000 km$^2$ area and further ranges between different regions of China, Afghanistan, Pakistan and India [24]. The portion of Indus upstream of the Tarbela Dam, is called Upper Indus Basin (UIB). UIB extends up to 1150 km in length with drainage area almost 165 400 km$^2$ [10]. Being a mountainous region, major portion of UIB is perennial glacial ice-covered area (approx. 15 062 km$^2$) with total estimated ice reserves of 2174 km$^3$ [25]. The altitude in the UIB ranges from 455 to 8611 m and climate varies greatly because of such altitudinal variability inside the catchment [16]. The brief description of the study area is shown in figure 1.

In UIB, 90% of catchment area is situated under the rain influence of Himalayas range, and therefore, the basin is least affected by summer monsoon [26]. In mountains towards the south, the incursion of the monsoon influence from the Indian Ocean is restricted, thus its effect fades northwestward [10,27]. As a result, the climate in the Indus Basin is somewhat varied when compared with the eastern side of Himalayas range. The annual precipitation in the UIB initiates in the western side, produced from the turbulences of western mid-latitude and happens typically during spring and winter [28–30]. In UIB,

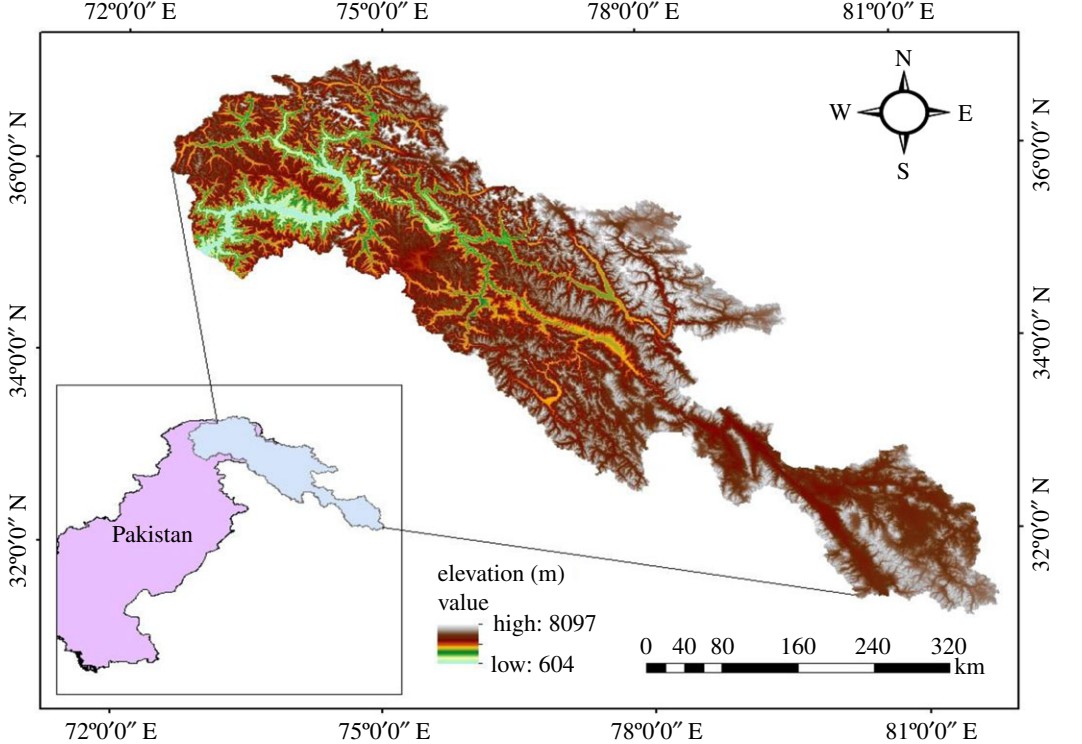

**Figure 1.** Study area (Upper Indus Basin) with digital elevation model.

the annual precipitation ranges from 100 to 200 mm specifically in the arid northern floors because of the strong influence of the topographic altitude on the climatic variables. The maximum snow accumulated (70–80%) during winter period (December to February), and the snow cover remains (10–15%) in melting time, i.e. June to September [16]. Similarly, flow in the UIB is a result of both glaciers melt from higher altitude and storm runoff in the lower parts [8] [24,31]. The flow to the whole Indus basin, 86–88% from summer and only 12–14% from winter is contributed by UIB [32,33] (table 1).

## 2.2. Datasets

### 2.2.1. Meteorological data

The study area comprised several meteorological stations. Out of total, six stations lie under jurisdiction of Pakistan Meteorological Department (PMD), while 14 stations are operated by Water and Power Development Authority (WAPDA), Pakistan. The PMD stations have climatic data on daily basis and available from 1947 to date but the data is inconsistent with huge gaps. However, fairly consistent data is available for the recent years. Because of low altitude, the PMD stations are not totally appropriate to the most parts of the UIB. The remaining stations of WAPDA were installed recently and also cover high altitude, but data for limited duration is available only for period 1999–2008. Due to the limitation of long-term data, most climate-change studies relied only on limited observation stations data. For this study, a long-term new gridded dataset [22] was used. The orographic effect in the region and particularly the problem induced because of higher elevations are successfully incorporated in the gridded data [22]. Details of all the datasets used are given in table 1.

### 2.2.2. Landcover data

Landcover data are the primary input data for SWAT modelling, as the variations in land use/land cover (LULC) could momentously affect runoff, evapotranspiration and certain other parameters of the hydrological cycle [34]. The dataset of landcover was acquired from GlobCover landcover product and it is the result of European Space Agency (ESA)-GlobCover. It is based on ENVISAT's MERIS Level 1B data with a spatial resolution of 300 m. The data was then clipped according to shape file in order to get map of the desired LULC for UIB. Based on LULC, the study area was divided into 16 major

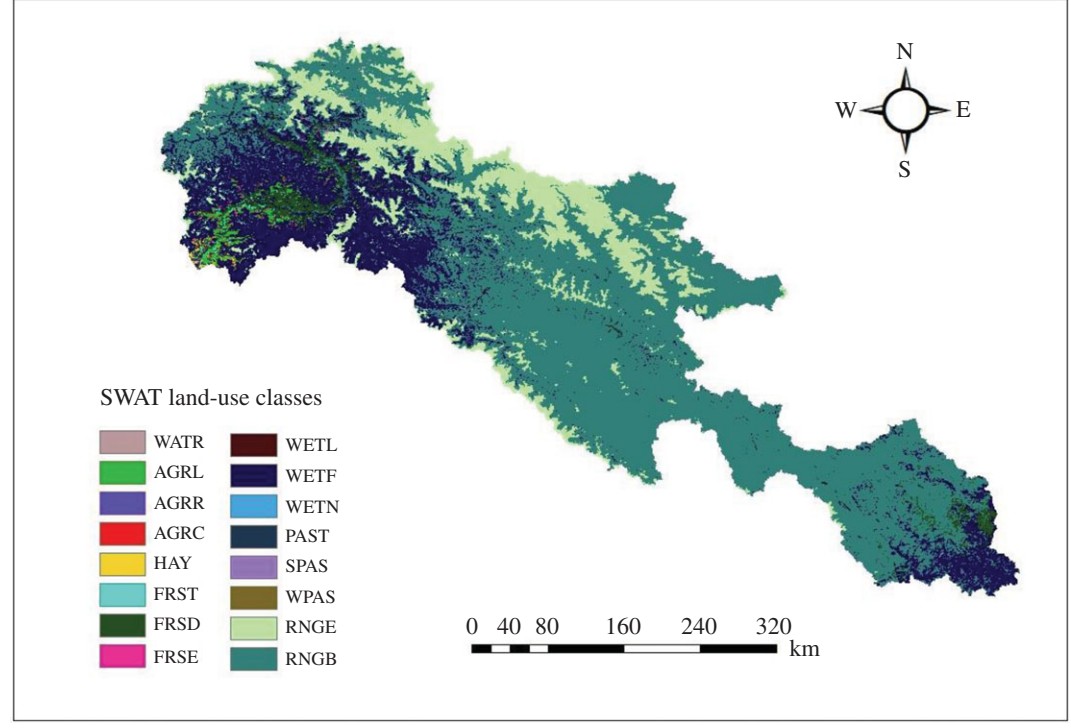

**Figure 2.** Land-cover classes observed in UIB.

**Table 1.** Source, description and type of data used in this study.

| data type | data source | scale | description |
|---|---|---|---|
| digital elevation model (DEM) | NASA-SRTM (30 m) | grid cell (30 × 30 m) | NASA Shuttle Radar Topographic Mission (SRTM) |
| land-use | GlobCover land cover product | 300 m | European Space Agency (ESA)-GlobCover land cover product |
| soil data | FAO/UNESCO soil data | 1 : 5 000 000 | FAO digital soil data |
| weather data | gridded dataset | daily data | long-term new gridded dataset [22] |
| river flow/ discharge data | surface water hydrology project of Water and Power Development Authority (WAPDA) (1975–2010) | daily data | mean daily discharge ($m^3 s^{-1}$) at Bisham Qilla gauge station |

categories (figure 2), which was then reclassified according to their hydrologic properties and SWAT requirement. For each land category, the model provides a unique four-letter code. The dominant classes were forest mixed, forest deciduous, range grasses and agricultural land-row crops etc.

### 2.2.3. Soil data

Different soil physico-chemical and textural properties are required for SWAT model such as available water content, soil texture, bulk density, hydraulic conductivity and carbon content for each layer of different soil types. For this study, FAO/UNESCO soil data were used with projection based on Universal Transverse Mercator (UTM) and 90 × 90 m resolution, and then applied in model for hydrological response unit (HRU) analysis. The FAO digital soil map of the world is the digitized

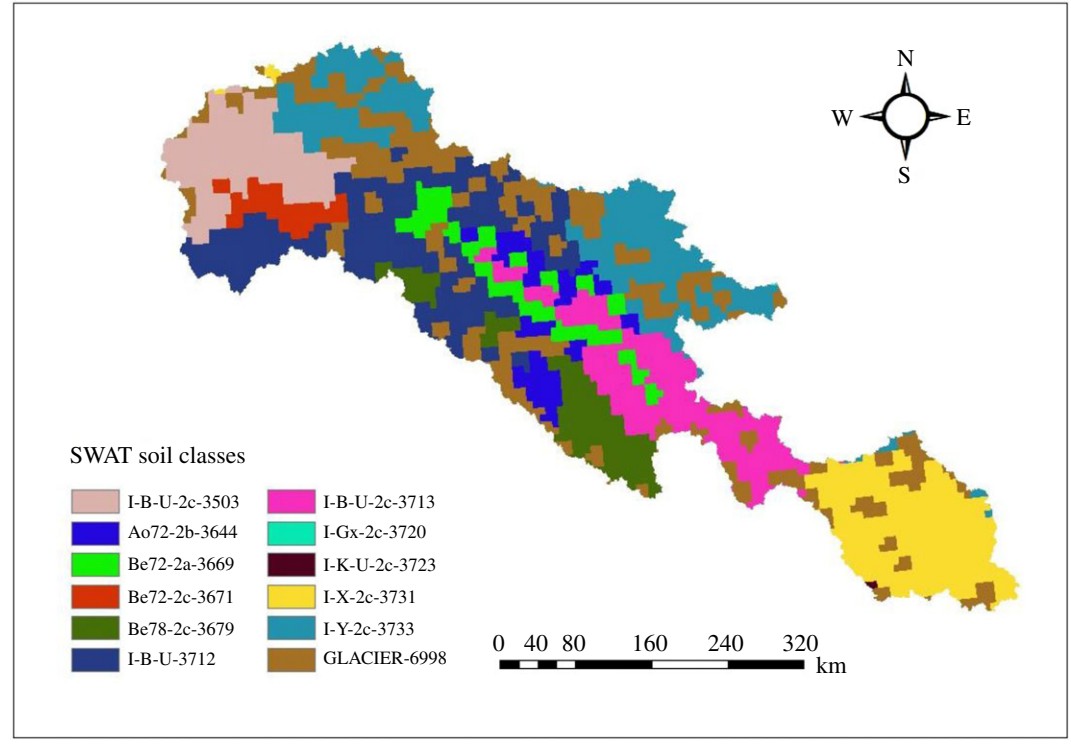

**Figure 3.** FAO soil classification (classes) in UIB.

**Table 2.** Description of the RCM used in the present study.

| s. no | experiment name | short form | driving AOGCM | RCM | RCM description |
|---|---|---|---|---|---|
| 1 | NorESM1-M_RCA4 | NOR | Nor-ESM1-M | RCA4 | Rossby Centre regional atmospheric model v. 4 (RCA4) [36] |

version of the FAO-UNESCO soil map of the world. After importing soil map to Arc-SWAT interface, 12 different soils were delineated in the basin (figure 3).

### 2.2.4. Regional circulation model base climate data

The climatic forecasts provided by general circulation models (GCMs) or regional circulation models (RCMs) might work as essential input data for climate change assessment. They need to be downscaled in order to meet the requirements of appropriate resolutions by means of statistical/dynamical downscaling or with the help of RCMs, set in a larger GCM [10]. The RCM projections are already available in coordinated regional climate downscaling experiment (CORDEX) and created fine-scale projections for different parts of the world, from which 'CORDEX-South Asia' experiments cover the UIB. From CORDEX, the RCM 'NorESM1-M_RCA4' was selected for the present study as it was found to be the most appropriate for depicting the median tendencies of future climate for UIB [35]. The description of the RCM used is given in table 2. Moreover, four RCPs could generally be used as a base mentioned in the IPCC fifth assessment report. RCP8.5 (very high baseline), RCP4.5 and RCP6 (medium base scenarios) and the moderated scenario (RCP2.6). The focus of this study was to consider RCPs similar to variety of future temperature variances and broader range of radiative forcing, and keeping the RCPs depicting much similarity with the 2005 onwards carbon dioxide emission growth rates and trend. RCP8.5 covers both the temperature change and extreme limit of radiative forcing. It also resembled the trend of about 3% annual (2005–2012) carbon dioxide emission rates [37,38]. As the medium case scenarios are concerned, both RCP6 and RCP4.5 were suitable; however, RCP4.5 showed similar trend

(1.5%) of average annual (2005–2012) $CO_2$ emission rates [37]. The low radiative forcing scenario (RCP2.6) was not considered. In the present period of industrialization, a drastic and immediate decrease in the emissions of greenhouse gases is very difficult; hence, it is unlikely to meet this mitigation scenario [17,39]. Hence, RCP4.5 was chosen along with RCP8.5, and the simulated RCM-based climatic variables ($T_{max}$, $T_{min}$ and precipitation) for the selected RCM were downloaded under RCP scenarios.

After selecting RCPs, the next phase was addressing the two prime issues constraining the climate studies: firstly, the output of GCMs/RCMs might not be fine as necessary for local-scale and regional studies, and secondly, the output from GCM/RCM was supposed to contain biases of definite extent, as compared with observed data. The data for the appropriate RCM was downloaded and the above-mentioned issues were addressed using bias correction by distribution mapping (DM) technique [40]. Using the DM method, the outputs of RCM were used for the two future time horizons, i.e. 2041–2070 (mid-century) and 2071–2100 (late-century) under RCP4.5 and RCP8.5 conditions.

# 3. Methodology

## 3.1. Description of soil and water assessment tool model

SWAT is a process-based long-term model specifically used for sediment, water movement and nutrients simulation on a watershed scale [41]. The purpose of the SWAT is to calculate the effect of sediment transport, water flow, crop growth, chemical/agricultural yields and nutrient cycling. Moreover, the interaction between climatic variables including surface water and land use could be studied by using this model [42]. The computational efficiency by dividing the watersheds into smaller sub-parts and capability of providing accurate spatial details makes the SWAT model more attractive and reliable. The model facilitates users by assessing the future scenarios using different input datasets such as land-use methods, water quality, climate, nutrient cycling, land cover, water movement and others to model watersheds [43]. The primary components of SWAT model include hydrology, weather, plant growth, soil temperature, land management nutrients and pesticides. Previously, all these components were confirmed and applied successfully for several watersheds [43].

### 3.1.1. Soil and water assessment tool snow module

In SWAT model, the average daily air temperature and boundary temperature are used as indicators to classify precipitation as rain or snow. If the mean daily air temperature could be below the boundary temperature, the precipitation is considered as snow. Similarly, if the temperature is above the boundary temperature, precipitation would be modelled in the form of liquid rain. Snowfall is stored at the ground surface in the form of accumulating snow pack, and the amount of water stored there is reported as snow water equivalent [41,44]. The mass balance of the snow pack was calculated using equation (3.1).

$$SNO = SNO + R_{day} - E_{sub} - SNO_{mlt}. \tag{3.1}$$

Where SNO is the water content of pack on a given day (mm $H_2O$), and $R_{day}$ is precipitation on a given day. $E_{sub}$ is the amount of sublimation on a given day (mm $H_2O$), and $SNO_{mlt}$ is the amount of snowmelt on a given day (mm $H_2O$). The influencing factors such as drifting, shading and irregular topography, the distribution of snow pack is not constant over the entire watershed [41]. In the present study, an aerial depletion curve was used to present the seasonal growth and decay of the snow pack as a function of the amount of snow present in the basin. This curve is based on a natural logarithm and was calculated using equation (3.2).

$$SNO_{cov} = \frac{SNO}{SNO_{100}} \times \left[ \frac{SNO}{SNO_{100}} + \exp\left( cov_1 - cov_2 \times \frac{SNO}{SNO_{100}} \right) \right]^{-1}. \tag{3.2}$$

The HRU area covered by snow is represented by $SNO_{cov}$, SNO is the water content of the snow pack on a given day (mm $H_2O$), $SNO_{100}$ is the threshold depth of snow at 100% coverage, and the shape of the curve is defined by coefficients $cov_1$ and $cov_2$. Furthermore, the snow pack temperature of the current day was calculated using the equation (3.3).

$$T_{snow(d_n)} = T_{snow(d_n-1)(1-1_{sno})} + \overline{T_{av}} \cdot 1_{sno}, \tag{3.3}$$

where $T_{snow(dn)}$ is the snow pack temperature on a given day (°C), $T_{snow(d_n-1)}$ is the snow pack

temperature on the previous day (°C), $1_{sno}$ is the snow temperature lag factor, and $T_{av}$ is the mean air temperature on the current day (°C).

### 3.1.2. Snow routing and temperature index approach

We used the temperature index approach [45] with elevation bands to incorporate the contribution of snow/ice melt in SWAT [17]. The temperature index algorithm allows the model to replicate the response of snow in glaciated watershed by distributing it to several elevation bands. The elevation bands are essential variables in hydro-meteorological parameters for both temperature indices and snow/ice quantities. SWAT model divides the sub-basin into 10 elevation bands, and then snow/ice melting is replicated separately for each elevation band [41], although the SWAT model assumes the glaciers area as static [46]. The temperature index approach was used by Babur *et al.* [15] in Jhelum River basin, Garee *et al.* [14] in Hunza River basin, Anjum *et al.* [17] in Swat River basin and Zhang *et al.* [47] in Yellow River basin for stream-flow simulation in glaciated watershed. In addition, the lapse rate approach was adopted from Khan & Koch [22] who used a new method for interpolation and correction of the data across the whole UIB [22]. In SWAT model, different methods could be used for evapotranspiration (ET) such as Hargreaves, Food and Agriculture Organization Penman–Monteith (FAO-PM) and Priestley–Taylor. In the current study, we used the Priestley–Taylor method that was found to be efficient based on initial model performance prior to calibration.

### 3.2. Model set-up

Initially, all the datasets used for modelling were projected under the same projection using Universal Transverse Mercator (UTM) Zone 43 N, hence they could be accurately overlaid and also calculations could be made easily [34]. DEM with resolution of $30 \times 30$ m obtained from NASA Shuttle Radar Topographic Mission (SRTM) was used and the whole watershed was segmented into 167 sub-basins. The LULC, soil and slope datasets were imported to the SWAT model. After successfully overlying the slope, soil and land-use datasets, the model generated 2778 HRUs with 100% overlapped with the watershed boundaries. The smallest unit of the basin is called hydrological response unit (HRU) in SWAT and is combination of soil, slope and LULC. The weather inputs contained the most important temperature and precipitation data. All the weather dataset records from 1975 to 2005 with 3 years of warm-up was used to get the effective hydrological cycle. Daily river discharge data of gauge station (Bisham Qilla) was acquired from WAPDA Pakistan from January 1960 to December 2005. The sensitivity analysis and model calibration and validation were accomplished using the obtained daily discharge data.

### 3.3. Sensitivity analysis and soil and water assessment tool calibration and validation process

The sensitivity analysis was the foremost step in finding the most sensitive parameters which required adjustment based on expertise and study area. There are a number of uncertainties related to model input parameter values. It is performed by varying the values of input parameters for the purpose to get better output, as it assesses the efficiency of the model and also helps to understand the performance of the system being modelled. In this study, the local sensitivity analysis (one at a time) was applied. The different sensitive parameters were identified which were important for flow, among which the most sensitive were selected after initial iteration run of model.

The calibration and validation process are essential for predicting the model efficiency. In this process, the model inputs are changed to attain the best agreement between the observed and simulated system variables. Using daily discharge data at Bisham Qilla Station, the model was first calibrated and then validated. The discharge data for a period 1978–1993 was used for calibration through SWAT-CUP v. 5.2.1.1. The sequential uncertainty fitting SUFI-2 algorithm of the SWAT-CUP program was used for calibration [48,49]. During this process, a specific range should be assigned to a set of parameters intended to be used for calibration, where both (calibration parameters and their ranges) are directed by precise knowledge of the study area, parameters sensitivity analysis, and literature information. The two specific indices, the P-factor and R-factor are used in terms of uncertainty levels for quantifying the model performance based on objective function set for the selected parameter ranges. The P-factor usually ranges between 0 and 1, where 1 indicates all the estimates/points inside the '95PPU' band, whereas, R-factor ranges from 0 to ∞, with 0 signifying complete match [48,50]. At last, the best fit values of the selected parameters were attained by setting 500 simulations in SWAT-CUP iterations. The validation was also executed to match the output of model with observed data without making adjustment in the parameter values. The discharge data from 1994 to 2003 was used for validation purpose.

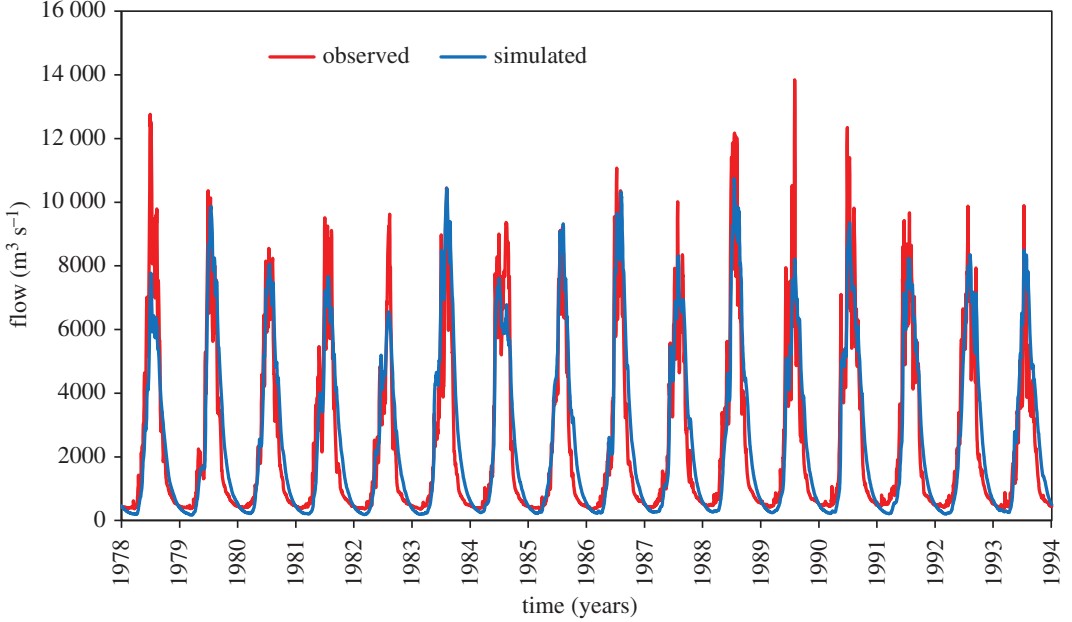

**Figure 4.** Flow calibration on daily basis for period 1978–1993.

## 3.4. Model performance evaluation

The SWAT model performance was measured using the statistical parameters comprising Nash–Sutcliff efficiency (NSE), coefficient of determination ($R^2$), Kling–Gupta efficiency (KGE) [51] and percent biasness (PBIAS) [47]. These parameters are used to check the simulated hydrological processes against the measured flow data. $R^2$ value lies between 0 and 1 and the higher value specifies the chances of less errors among the simulated and observed records, while value 1 represents the regression line exactly according to the data and shows 100% matching of simulated and observed record [52]. NSE value lies between negative infinity and 1 (perfect), where NSE value equals 1 shows perfect match. If NSE is greater than 0.65, the correlation is reflected as very good [52]. Negative NSE value points out very poor results; it means that average value of output is a better approximation than the model forecast [53]. The KGE is a parameter based on NSE and mean squared error. It shows the correlation, relative variability and bias between observed and simulated values [51]. Like NSE, the KGE ranges from -∞ to 1 with the optimal value as 1 [51,54]. PBIAS is used to estimate the efficiency of simulated data from observed data and expressed as percentage. The percent difference calculates the mean difference between measured and simulated data over a definite period. Typically, PBIAS ranges from −20 to 20. The positive biasness indicates model underestimation, whereas negative values represent model overestimated results [55].

# 4. Results

## 4.1. Model calibration and validation

The calibration was completed utilizing daily flow data of UIB at Bisham Qilla station from 1975 to 1993. During this process, sequential uncertainty fitting (SUFI-2) algorithm of the SWAT-CUP program was used. The daily observed and simulated datasets were plotted to find out the model efficiency by using NSE, $R^2$ and PBIAS. The flow calibration results for UIB exhibited an excellent correlation between observed and simulated values. Description of the statistical indicators are shown in table 3. During calibration, mean daily flow for simulated and observed data were found to be 2604.13 and 2442.80 m$^3$ s$^{-1}$ respectively. Similarly, for validation period, average daily flow for simulated data was 2659.15 and 2369.12 m$^3$ s$^{-1}$ for the observed data, which showed a very good match.

The correlation between observed and simulated data is graphically presented in figure 6. For calibration results, both the NSE and $R^2$ were found as 0.85 for daily flow. Similarly, PBIAS was observed as −6.6. The statistical value 0.85 shows that the outcome predicted by the model is reliable. The calibration graph is presented in figure 4. Similarly, the efficiency of calibrated model was further validated from 1994 to 2003 on daily flow data. The model performance evaluation criteria fulfilled the requirement of NSE >

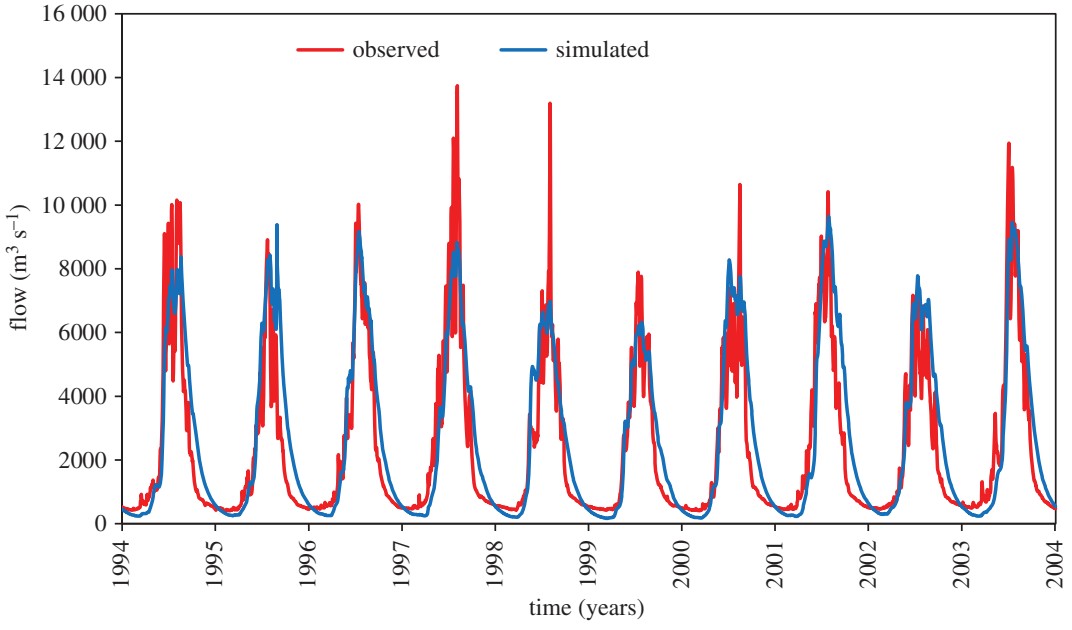

**Figure 5.** Flow validation on daily time step from 1994 to 2003.

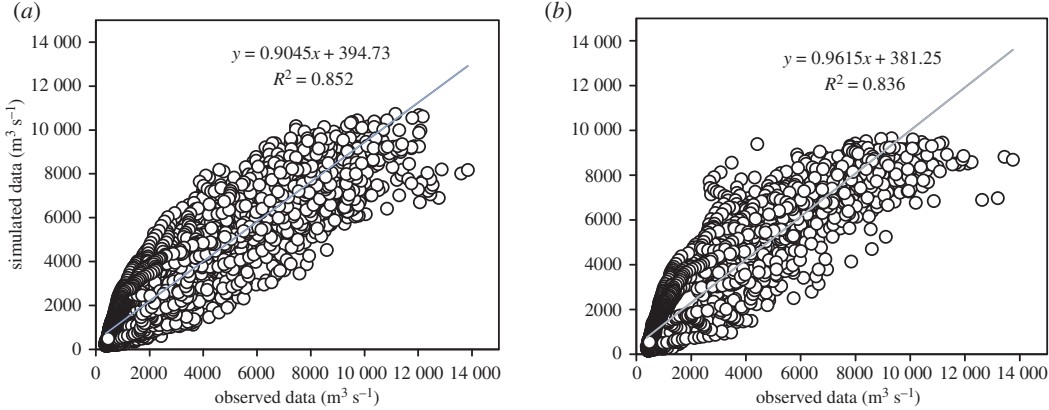

**Figure 6.** Correlation between observed and simulated flow (*a*) calibration period (*b*) validation period.

**Table 3.** Statistical parameters for observed and simulated flow during calibration and validation.

| coefficients | calibration period (1978–1993) | | validation period (1994–2003) | |
|---|---|---|---|---|
| | simulated | observed | simulated | observed |
| average flow | 2604.13 m$^3$ s$^{-1}$ | 2442.80 m$^3$ s$^{-1}$ | 2659.15 m$^3$ s$^{-1}$ | 2369.12 m$^3$ s$^{-1}$ |
| p-factor | 0.76 | | 0.78 | |
| r-factor | 0.81 | | 0.79 | |
| $R^2$ | 0.85 | | 0.84 | |
| NS | 0.85 | | 0.80 | |
| PBIAS | −6.6 | | −12.2 | |
| KGE | 0.90 | | 0.84 | |

0.5 and $R^2 > 0.6$ proposed by SWAT developer [56]. In validation, the $R^2$ and NSE for daily flow were found to be 0.84 and 0.80 respectively. Also, the PBIAS was found to be −12.2. The validation results for the period 1994–2003 are shown in figures 5 and 6 and tables 3 and 4

**Table 4.** Most sensitive parameters used during calibration and validation process.

| parameter name | description | range |
|---|---|---|
| SOL_AWC.sol | available water capacity of the soil layer | 0–1 |
| PLAPS.sub | precipitation lapse rate | −1000–1000 |
| HRU_SLP.hru | average slope steepness | 0–1 |
| GW_SPYLD.gw | specific yield of the shallow aquifer ($m^3\ m^{-3}$) | 0–0.4 |
| SUB_SMFMN.sno | minimum melt rate for snow during the year | 0–20 |
| OV_N.hru | Manning's 'n' value for overland flow | 0.01–30 |
| REVAPMN.gw | threshold depth of water in the shallow aquifer for 'revap' to occur (mm) | 0–500 |
| SMTMP.bsn | snowmelt base temperature | −20–20 |
| CH_N1.sub | Manning's 'n' value for the tributary channels | 0.01–30 |
| SUB_SMTMP.sno | snowmelt base temperature | −20–20 |
| SUB_TIMP.sno | snow pack temperature lag factor | 0–1 |
| SMFMX.bsn | maximum melt rate for snow during year | 0–20 |
| CH_S1.sub | average slope of tributary channels | 0.0001–10 |
| CH_S2.rte | average slope of main channel | −0.001–10 |
| GW_REVAP.gw | groundwater 'revap' coefficient | 0.02–0.2 |
| CH_K1.sub | effective hydraulic conductivity in tributary channel alluvium | 0–300 |
| SMFMN.bsn | minimum melt rate for snow during the year | 0–20 |
| SUB_SFTMP.sno | snowfall temperature | −20–20 |
| TIMP.bsn | snow pack temperature lag factor | 0–1 |
| SFTMP.bsn | snowfall temperature | −20–20 |
| SNOCOVMX.bsn | minimum snow water content that corresponds to 100% snow cover | 0–500 |
| SURLAG.bsn | surface runoff lag time | 0.05–24 |
| CN2.mgt | SCS runoff curve number | (0–100)% |
| CH_N2.rte | Manning's 'n' value for the main channel | −0.01–0.3 |
| SLSUBBSN.hru | average slope length | 10–150 |
| CH_K2.rte | effective hydraulic conductivity in main channel alluvium | −0.01–500 |
| ALPHA_BNK.rte | base-flow alpha factor for bank storage | 0–1 |
| TLAPS.sub | temperature lapse rate | −10–10 |
| SOL_K.sol | saturated hydraulic conductivity | 0–2000 |
| SNO50COV.bsn | snow water equivalent that corresponds to 50% snow cover | 0–1 |
| ALPHA_BF.gw | base-flow alpha factor (days) | 0–1 |
| GW_DELAY.gw | groundwater delay (days) | 0–500 |
| SUB_SMFMX.sno | maximum melt rate for snow during year | 0–20 |
| SLSOIL.hru | slope length for lateral subsurface flow | 0–150 |

The observed sensitive parameters during calibration and validation are presented in table 4. The parameters related to soil moisture used in calibration are SOL_AWC and SOL_K. The parameter SOL_AWC represents the available moisture content in soil between wilting point and field capacity. Hydraulic conductivity of soil layers with water content could be adjusted by using SOL_K parameter [57]. As the UIB is a highly glaciered basin, the glacier and snowmelt parameters were the most important and affected the model performance. The different parameters related to and the most sensitive to glacier/snowmelt are SMFMN, SMTMP, TIMP, SMFMX, SFTMP, SNO50COV and SNOCOVMX. The SFTMP (snowfall temperature) was used as a threshold to categorize the precipitation as rainfall or snowfall. When the air temperature is lower than SFTMP, the precipitation is classified as snowfall [41]. The snow pack would not melt until the snow pack temperature exceeds the threshold value i.e. SMTMP (snowmelt base temperature). SMFMX is the melt factor for 21 June and SMFMN is

**Table 5.** Probable changes in temperature and precipitation under both RCPs for mid and late century.

| duration | RCP4.5 | | RCP8.5 | |
|---|---|---|---|---|
| | temperature (°C) | precipitation (%) | temperature (°C) | precipitation (%) |
| 2041–2070 | 2.36 | 2.4 | 2.92 | 6.0 |
| 2071–2100 | 3.5 | 2.5 | 5.23 | 4.6 |

the melt factor for 21 December [41]. The SNOCOVMX represents the snow water content agreeing with watershed full snow cover [14,58]. During melting season, the snow cover and associated snow volume affects the actual meltwater discharge. In a melting season, a reduction in snow meltwater might be used to correctly calculate actual melting volume from snow [14]. The quantity of water staying on ground surface can be predicted by means of depletion curve which is fixed prior to snow/glacier melt. The SNO50COV parameter which is volume of snow water that relates to 50% snow cover, could be used to adjust shape of depletion curve [59]. The SNO50COV ranges from 0 to 1, and, based on study area, the shape of the curve could be adjusted at several positions. The TIMP is an empirical parameter used for snow density, extent, depth and other factors affecting snow pack temperature [58]. TIMP is used to regulate temperature effect on previous days based on current temperature of snow pack.

The contribution of ground water was adjusted by different parameters, out of which GW_DELAY and ALPHA_BF were the most important. GW_DELAY represents the delayed time when water enters from soil layer to shallow aquifer. Decreasing GW_DELAY value affected both the peak discharge and quantity of available base-flow water [57]. ALPHA_BF is the coefficient used for changes in ground water flow related to changes in recharge [41]. A decreasing ALPHA_BF could be linked to that recharge to aquifer is slow and resulting in low annual flow in melting season, but for future stream flow, the amount of stored water is increasing.

## 4.2. Climate change analysis

### 4.2.1. Projected change in temperature and precipitation

The output of RCM was corrected for biases with the help of distribution mapping under two representative concentration pathways (RCP4.5 and RCP8.5) and used further to assess future climate, i.e. temperature and precipitation. The estimated changes in precipitation and temperature under RCP4.5 and RCP8.5 for both mid and late century are shown in table 5. For RCP4.5, the projected change in temperature varies from 2.36° C to 3.50°C, and the change ranges from 2.92°C to 5.23°C for RCP8.5 till the end of the twenty-first century. Similarly, the forecasted changes in precipitation vary from 2.4% to 2.5% for RCP4.5 and 4.6% to 6% under RCP8.5. Akhtar *et al.* [60] used two HBV models i.e. HBV-met and HBV-PRECIS for climate change assessment. The PRECIS RCM data used showed an increase of 4.88°C in mean annual temperature in the UIB till the end of the twenty-first century. Forsythe *et al.* [61] used a stochastic rainfall model and projected an increase in precipitation by 27% seasonal and 18% mean annual changes in UIB. The above-cited studies are in line with our findings and suggested a projected increase in temperature as well as precipitation. Therefore, as our findings, both temperature and precipitation increased in UIB under both RCPs and revealed rise in the average annual flows.

### 4.2.2. Projected increase in stream flow

Using data corrected for biases, for the CORDEX RCM model NorESM1-M_RCA4, the model was run for two time lapses, i.e. 2041–2070 and 2071–2100. For individual case, the future hydrology was assessed for annual average for two periods. The calculations were based on percent relative changes from the reference period (1976–2005). The results are summarized in table 6 and graphically shown in figures 7–9. From results, it is clear that the projected variations in flow fairly showed a steady increase in both mid and late centuries for all future climate scenarios as compared with baseline period (1976–2005). In RCP4.5, comparing with reference period, the projected increase in mean annual flow is 19.24% and 16.78% for mid and late century respectively. Similarly, this increase in mean annual flow is 20.13% and 15.86% under RCP8.5 for period 2041–2070 and 2071–2100 respectively. Furthermore, under both RCPs, the increase in average annual flow would be high (19.24% to 20.13%) during mid-century (2041–2070), whereas the flow increase would be less (15.86%

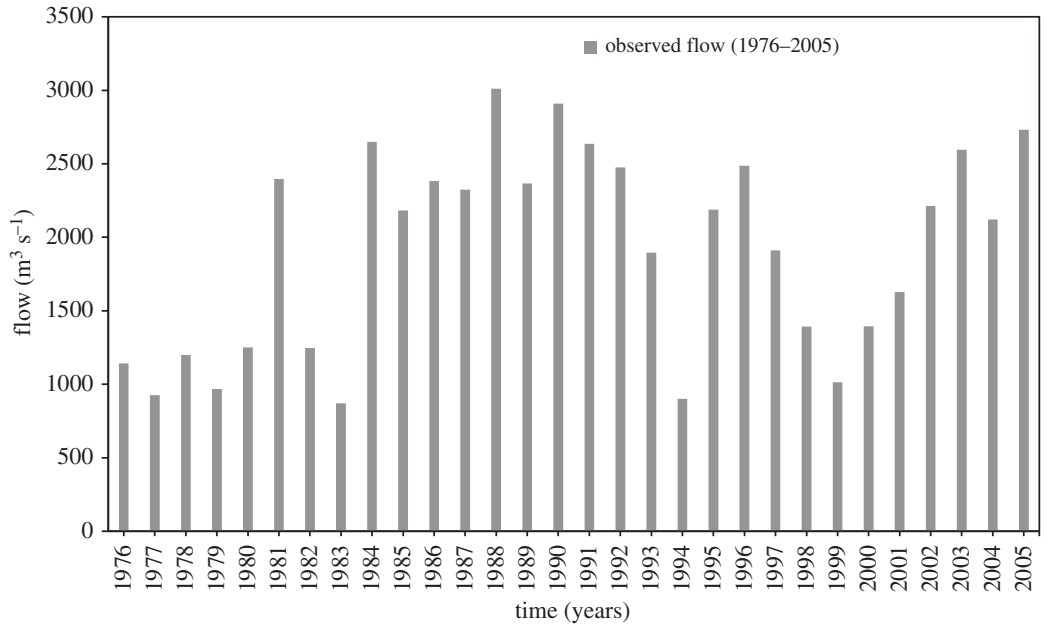

**Figure 7.** Observed average annual flow for the baseline period (1976–2005).

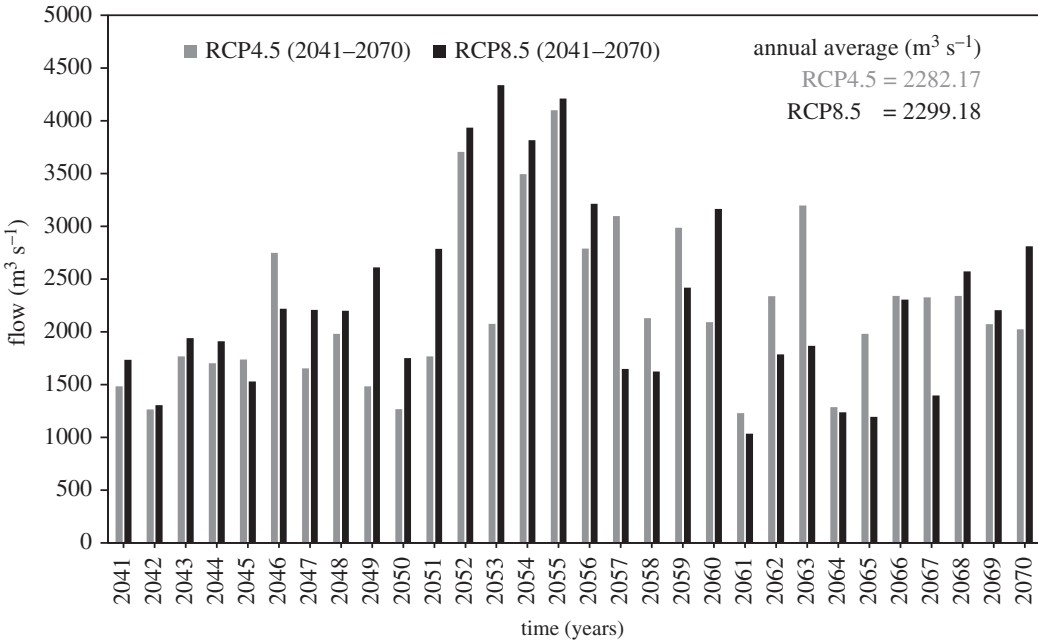

**Figure 8.** Mid-century (2041–2070) flow comparison under RCP4.5 and RCP8.5.

**Table 6.** Estimated variation in flow in both mid and late century under RCP4.5 and RCP8.5.

| mean annual observed flow (1976–2005) = 1913.87 m³ s⁻¹ | | |
|---|---|---|
| duration | RCP4.5 value (change %) | RCP8.5 value (change %) |
| 2041–2070 | 2282.17 (+19.24) | 2299.18 (+20.13) |
| 2071–2100 | 2235.05 (+16.78) | 2217.55 (+15.86) |

to 16.78%) during late century (2071–2100). Likewise, in mid-century, the simulation result shows more variation in flow under RCP8.5 when compared with RCP4.5. In the late century (2071–2100) under RCP8.5, the flow could increase by 15.86%. In the same period (2071–2100), flow could increase by 20.13% under RCP4.5.

## 5. Discussion

In the previous section, the expected changes in temperature, precipitation and stream flow under RCP4.5 and RCP8.5 for both mid and late twenty-first century were summarized in detail. Based on our results, high average annual flow was observed in mid-century when compared with late century. The lower overall flow magnitudes over the late century (2071–2100) in comparison to the flow in mid-century (2041–2070) could be attributed to the decrease in meltwater contributions over time [20] and possibly to the elevated rates of evapotranspiration with the temperature rise, offsetting any increase in contribution from precipitation or meltwater [20,62]. Moreover, during mid-century, the climate forcing under RCP8.5 produced flows with higher variation, when compared with that under RCP4.5. This could be due to the increase in: (i) radiative forcing, (ii) the expected precipitation, and (iii) the temperature for RCP8.5. The outcome is higher melt and direct runoff contributions, resulting in increase in mean annual flow [37,62].

Similarly, the projected flows for the late century (2071–2100) also showed interesting results for RCP4.5 and RCP8.5, where the projected flow was higher under RCP4.5 in comparison to RCP8.5. In fact the projected flows under RCP4.5 showed an increase of 19.24% and 16.78% in mean annual flow for mid (2041–2070) and late (2071–2100) twenty-first century, respectively, while for RCP8.5, this increase in mean annual flow was observed to be 20.13% and 15.86% for mid and late century, respectively. In the case of both RCPs, the late century witnesses lower mean annual flows in comparison to the mid-century despite higher temperatures that could induce higher melting. This is due to the fact that, in the late-century, the increased contribution to the flows by snowmelt (due to higher temperatures) is probably balanced out and offset by the reduction of snowfall and thus accumulation [10] and the increased rates of evapotranspiration, which acted as a further loss from the water balance and resulted in reduced annual flows under RCP8.5 in late century (2071–2100) [37,63].

Shrestha & Nepal [18] reported that out of the total discharge, 45% and 47% was contributed by snowmelt and glacier ice melt respectively from Hunza basin. Shrestha *et al*. [8] revealed that runoff in the Hunza River was strongly influenced by the snow and glacier melt with almost 50%, 33% and 17% contribution from snow, glacier melt and rainfall, respectively. Tahir *et al*. [16] used the snowmelt-runoff model (SRM) along with MODIS remote sensing snow-cover data. The model results under future climate revealed almost doubled summer runoff till the mid-century. Wijngaard *et al*. [20] concluded that the increased flow might be attributed mainly to the increased precipitation and the temperature extremes. Chevallier [64] showed that the stream flow in the Hunza River is mainly subjected to winter precipitation and also to the mean summer and winter temperatures. The change in snow cover was inversely related to summer mean temperatures, resulting in an increased stream flow. Hayat *et al*. [7] used SRM in Astore and Hunza basin and predicted an increase of 13–58% and 14–90% stream flow for Astore and Hunza basin, respectively. Garee *et al*. [14] predicted that annual surface flow could increase ranging from 7.57% to 32.12% till the end of twenty-first century with a temperature increase from 1.39°C to 6.58°C and precipitation increase up to 31%. Anjum *et al*. [17] used SWAT in SWAT river basin and directed that the temperature increased up to 4.18°C and 8.49°C and precipitation rose by 22.52% and 35.98% for RCP4.5 and 8.5, respectively.

The above-cited studies have similarities to our results. It is probable that the temperature increase, projected precipitation and accelerated melt rates from snow/glaciers are the key factors responsible for future hydrology in the UIB. The main source for climate change study might be the selection and structure of proper GCM/RCM, and also the type of hydrological model produces uncertainty in climate studies [14].

## 6. Conclusion

In this research, the SWAT model was efficiently applied to observe the climate change impact on stream flow in UIB at the outlet of Bisham Qilla station. The results provided a valuable insight about runoff and water quantity in the future and are important for water resource managers and agencies to minimize their effect.

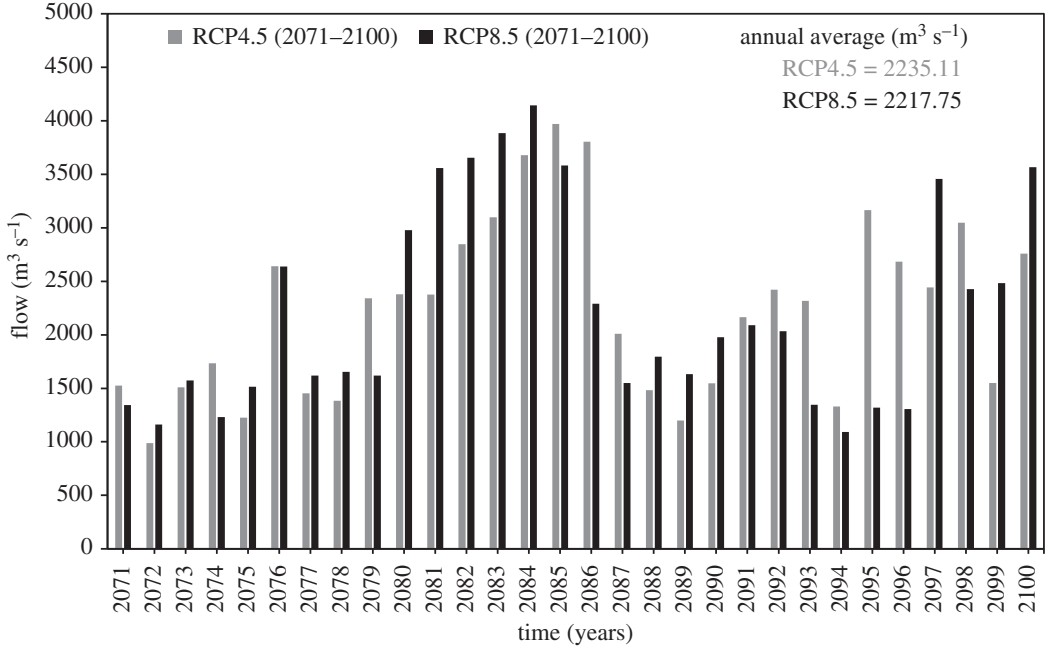

**Figure 9.** Late century (2071–2100) flow comparison under RCP4.5 and RCP8.5.

The SWAT model works in the framework of GIS to generate outputs of hydrological components based on GIS and meteorological input datasets. The (CORDEX-South Asia) RCM model NorESM1-M_RCA4 was selected for the study and used under two RCPs (4.5 and 8.5) for two time intervals, i.e. mid-century (2041–2070) and late-century (2071–2100) to estimate changes in precipitation, temperature and hydrological components in the study area.

The model was found to be efficient in the simulation of watershed hydrology and the calibrated model satisfactorily simulated stream flow with agreeable statistical parameters. In general, the daily stream-flow data generated by SWAT model matched well with the measured data. The value of statistical indicators $R^2$ and NSE for daily flow calibration was 0.85 each. During validation, $R^2$ and NSE were found to be 0.84 and 0.80, respectively, and showed a good fit for the data. Based on the results of the climate scenarios for two representative concentration pathways (RCP4.5 and RCP8.5) the following results were concluded:

— A warmer climate could be expected for the study area in future with a projected change in temperature between 2.36°C and 3.5°C under RCP4.5 (from mid-century to late century) and 2.92°C to 5.23°C under RCP8.5 (from mid-century to late century).
— Likewise, the anticipated changes in precipitation ranges from 2.4% to 2.5% for RCP4.5 and 6% to 4.6% under RCP8.5 (from mid-century to late century).
— The application of SWAT model under climate change projection suggested an increase of 19.24–16.78% in mean annual flow for mid (2041–2070) to late (2071–2100) twenty-first century, respectively, for RCP4.5. This increase in mean annual flow was observed to be 20.13–15.86% for RCP8.5 for mid to late century, respectively.
— The rise in river flow was high during mid-century when compared with late century. This is due to the fact that, in the late century, the increased contribution to the flows by snowmelt (due to higher temperatures) could be balanced out and offset by the reduction of snowfall and thus accumulation [10] and the increased rates of evapotranspiration, which acted as further loss from the water balance and resulted in reduced annual flows under RCP8.5 in late century [10].

The results indicated that if the climate change projections come true, an increasing flow under all RCP scenarios could be generated until the end of the twenty-first century. Rising temperature might lead to increased snow and ice melting, which could increase the intensity and frequency of floods in the future. According to the present study, an effective plan might be needed for better and sustainable management of water resources in future. Moreover, there is a need to incorporate a fully dedicated glacier module or coupling of a glacier model with SWAT for study of glaciers and their response in the catchment.

Ethics. This research study does not require any ethical approval.

Data accessibility. All datasets relevant to this study are uploaded and available online at the Dryad Digital Repository: https://doi.org/10.5061/dryad.1ns1rn8q8 [65].

Authors' contributions. M.I.S., A.K., T.A.A., Q.K.H., A.J.K. and A.D. contributed in the conceptualization of the study; M.I.S. and A.J.K. contributed in data analysis; A.K. and T.A.A supervised this research, M.I.S. drafted the original manuscript; A.K., T.A.A., Q.K.H., A.J.K. and A.D. contributed in reviewing and editing the manuscript.

Competing interests. The authors declare competing interests. At the time of writing, Prof. Quazi Hassan was a Board Member of Royal Society Open Science, but had no involvement in the review or assessment of the paper.

Funding. This research has no source of funding from any funding agency.

Acknowledgments. The authors acknowledge the support of Water and Power Development Authority (WAPDA) for providing hydrological data. The authors also acknowledge NASA, (ESA)-GlobCover and FAO/UNESCO for providing access to DEM, land-use and soil data, respectively.

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
