## [Reviewer comments · Royal Society Open Science]

Review History

RSOS-191957.R0 (Original submission)

Review form: Reviewer 1

Is the manuscript scientifically sound in its present form?

Yes

Are the interpretations and conclusions justified by the results?

Yes

Is the language acceptable?

No

Do you have any ethical concerns with this paper?

No

Have you any concerns about statistical analyses in this paper?

No

Recommendation?

Accept with minor revision (please list in comments)

Comments to the Author(s)

See attachment (Appendix A).

Review form: Reviewer 2

Is the manuscript scientifically sound in its present form?

Yes

Are the interpretations and conclusions justified by the results?

Yes

Is the language acceptable?

Yes

Do you have any ethical concerns with this paper?

No

Have you any concerns about statistical analyses in this paper?

No

Recommendation?

Major revision is needed (please make suggestions in comments)

Comments to the Author(s)

The method part is not clear how the glacier dynamics has been taken into account. The authors can also compare the results with other published literature. Also, the authors have missed major publications from the upper Indus basin hydrology. I have provided some feedback (Appendix B).

Decision letter (RSOS-191957.R0)

04-Feb-2020

Dear Dr Akbar,

The editors assigned to your paper ("Predicting Hydrologic Responses to Climate Change in Highly Glacierized and Mountainous Region Upper Indus Basin") have now received comments from reviewers. We would like you to revise your paper in accordance with the referee and Associate Editor suggestions which can be found below (not including confidential reports to the Editor). Please note this decision does not guarantee eventual acceptance.

Please submit a copy of your revised paper before 27-Feb-2020. Please note that the revision deadline will expire at 00.00am on this date. If we do not hear from you within this time then it will be assumed that the paper has been withdrawn. In exceptional circumstances, extensions may be possible if agreed with the Editorial Office in advance. We do not allow multiple rounds of revision so we urge you to make every effort to fully address all of the comments at this stage. If deemed necessary by the Editors, your manuscript will be sent back to one or more of the

original reviewers for assessment. If the original reviewers are not available, we may invite new reviewers.

- Data accessibility

If you wish to submit your supporting data or code to Dryad (<http://datadryad.org/>), or modify your current submission to dryad, please use the following link:
<http://datadryad.org/submit?journalID=RSOS&manu=RSOS-191957>

- Competing interests

- Authors' contributions

AB carried out the molecular lab work, participated in data analysis, carried out sequence alignments, participated in the design of the study and drafted the manuscript; CD carried out

the statistical analyses; EF collected field data; GH conceived of the study, designed the study, coordinated the study and helped draft the manuscript. All authors gave final approval for publication.

- Acknowledgements

- Funding statement

on behalf of Dr Mark Smith (Associate Editor) and R. Kerry Rowe (Subject Editor)
openscience@royalsociety.org

Associate Editor's comments (Dr Mark Smith):

Thank you for submitting "redicting Hydrologic Responses to Climate Change in Highly Glacierized and Mountainous Region Upper Indus Basin" to Royal Society Open Science. I have received two reviews of your manuscript, which are included below and/or attached. Both reviews indicate that this is an interesting and generally well implemented modelling study; however, one reviewer highlights several areas that require further attention. My own views on the paper are well-aligned with this review.

While I suggest that you address each of the comments indicated in the review, a greater description of the treatment of glacier melt and further discussion of the findings (especially in reference to previous work in the Indus basin) is especially important.

I am therefore returning the paper to you so that you can make the necessary changes, which fall under the category of 'Major Revisions'.

Reviewers' Comments to Author:

Reviewer: 1

Comments to the Author(s)

See attachment ("review_comments.pdf")

Reviewer: 2

Comments to the Author(s)

The method part is not clear how the glacier dynamics has been taken into account. The authors can also compare the results with other published literature. Also, the authors have missed major publications from the upper Indus basin hydrology. I have provided some feedback. see attachment ("reveiwer comments_RSOS_191957_SantoshNepal2.pdf")

Author's Response to Decision Letter for (RSOS-191957.R0)

See Appendices C & D.

Decision letter (RSOS-191957.R1)

Dear Dr Akbar:

On behalf of the Editors, I am pleased to inform you that your Manuscript RSOS-191957.R1 entitled "Predicting hydrologic responses to climate changes in highly glacierized and mountainous region Upper Indus Basin" has been accepted for publication in Royal Society Open Science subject to minor revision in accordance with the Editor suggestions.

The Editors have recommended publication, but also suggest some minor revisions to your manuscript. Therefore, I invite you to respond to the comments and revise your manuscript.

- Ethics statement

- Data accessibility

If you wish to submit your supporting data or code to Dryad (<http://datadryad.org/>), or modify your current submission to dryad, please use the following link:
<http://datadryad.org/submit?journalID=RSOS&manu=RSOS-191957.R1>

- Competing interests

- Authors' contributions

- Acknowledgements

- Funding statement

Because the schedule for publication is very tight, it is a condition of publication that you submit the revised version of your manuscript before 23-Jul-2020. Please note that the revision deadline will expire at 00.00am on this date. If you do not think you will be able to meet this date please let me know immediately.

on behalf of Dr Mark Smith (Associate Editor) and R. Kerry Rowe (Subject Editor)
openscience@royalsociety.org

Associate Editor Comments to Author (Dr Mark Smith):

Associate Editor: 1

Comments to the Author:

Dear Muhammad Shah and co-authors,

My apologies for the delay in returning these comments to you. As already communicated, we had problems in seeking a sufficient number of referee reports, no doubt at least in part a consequence of the current global pandemic.

Many thanks for providing a rebuttal to the major corrections suggested by the first original reviewer. I have now assessed these myself. Overall, these have improved the clarity and flow of the manuscript and helped place the study more clearly alongside existing research on the topic. I have the following comments, based on the original 'major changed' suggested:

- 1) The introduction and discussion now provide greater context and links with existing literature. The comparison with other work (P14) is especially informative. As a minor addition, I think it would be useful to include a sentence towards the end of section 1, indicating clearly how this study builds upon those existing studies cited. Essentially, a clear statement of novelty would be useful here.
- 2) Details of the methods applied are now much clearer (though see my comment below). Justifications for research choices (e.g. choice of RCM) are now provided and the precedents for these research methods stated explicitly, which is good to see.
- 3) I think there remains a need for greater clarity around the representation of glaciers in the model. The reviewer rightly highlighted this as a point requiring further description, given the expected significance of glacier melt in the future hydrology of the study basin. The reviewer questioned a number of times whether or not the glaciers are represented dynamically in the model, and I remain unsure of the answer even having read through the new corrections. I think this point needs to be addressed prior to publication.
- 4) All minor comments have been addressed.
- 5) The writing is clear and understandable throughout, but certainly a final proof read is required. This most likely comes at a later stage in the publication process, but I thought it best to raise as a note here.

Based on the above, I suggest that only minor changes are needed before this paper should be published in RSOS. These should not be too onerous and do not require any new data analysis.

- Add a clear statement of novelty (point 1 above)

- Address the issue of the treatment of glaciers in the model (point 3 above). You do provide examples of this same approach being used elsewhere, so I do not see a major issue here.

However, given the importance of glacier changes on the hydrology of the basin, comparison

with other studies (that may have used a more dynamic treatment of glaciers?) would be useful. Perhaps even a conceptual flow chart as to how glacier response is incorporated would be useful. I see this as the main outstanding issue here, but should be relatively easy to address.

Associate Editor: 2

Comments to the Author:

Many thanks for revising the manuscript as detailed. Given the nature of the changes and addition of whole sections of text, I feel it appropriate to send this out to review.

Author's Response to Decision Letter for (RSOS-191957.R1)

See Appendix E.

Decision letter (RSOS-191957.R2)

Dear Dr Akbar,

It is a pleasure to accept your manuscript entitled "Predicting hydrologic responses to climate changes in highly glacierized and mountainous region Upper Indus Basin" in its current form for publication in Royal Society Open Science.

Kind regards,

Andrew Dunn

on behalf of Dr Mark Smith (Associate Editor) and R. Kerry Rowe (Subject Editor)
openscience@royalsociety.org

Appendix A

Review of Predicting Hydrologic Responses to Climate Change in Highly Glacierized and Mountainous Region Upper Indus Basin by Shah et al.

This manuscript documents an application of climate downscaling to upper Indus basin. The problem setup is generally Ok, but writing needs to be significant improved. Currently there are many typos and grammar errors. I only give several examples below

P3, L15, "...severe storms and floods have ~~been~~ occurred in its history" should be "severe storms and floods have occurred in its history"

P3, L16-17, "Besides, the swift population growth and the accompanying land-use alterations further intensifying the problems induced by climate-change ..." should be "Besides, the rapid population growth and the accompanying land-use alterations further intensify the problems induced by climate-change"

P4, L10-L11, "The River Indus can be classifies as one of the major river in Asia,..." => "The Indus River can be classified as one of the major rivers in Asia,"

Appendix B

Royal society of open access

Predicting Hydrologic Responses to Climate Change in Highly Glacierized and Mountainous Region Upper Indus Basin

Muhammad Izhar Shah, Asif Khan, Tahir Ali Akbar, Quazi K Hassan, Asim Jahangir Khan and Ashraf Dewan

The paper looked at the impact of climate change on river flows in the Upper Indus basin using SWAT hydrological model which was forced with high resolution regional climate model (RCMs) for two future scenarios RCP4.5 and 8.5. The model has able to replicate the historic reference period well, based on which the future hydrology is quantified. Under different scenarios, the future flows are likely to increase.

Major comments

- The authors fail to cite important publications on the hydrological aspect of the Upper Indus Basin. I have listed a few important here (but I encourage authors to look more)
 - Shrestha, S., Nepal, S (2019)....
 - Shrestha Maheshwar (2015).....
- I think it would be very useful if authors compare the results with these published results to see where the historical hydrological modelling from this study compares
- It is not clear how the glacier response in the future in the model? Is glacier area changes due to the response of temperature change in the future (eg. Wijngaard et al. 2017). Shrestha and Nepal, 2019 also showed the different response to streamflow under various glacier recession scenarios.
- The methodology of using one RCM from CORDEX is not strong methodology, as the bias within one model could be high. There are many studies which look at the ensemble approach of few models.
- The increase in future flow is attributed to what? Increase in rainfall or increase in accelerated melting from glacier. It is not clear how the future temperature has affected glacier response. Is the SWAT glacier incorporation (if yes) is dynamic (changes in glacier area in the future) and stagnant?
- It is not clear how the authors have implemented glacier melt in the SWAT model. Authors have used the snow/glacier melt but it is not clear whether snow and glacier melt are deal in a different way, as both have different melt processes; snow is more seasonal and based on available snow storage, but glacier remains available even the snow is gone.
- Page 10, Line 35: Table 3: None of the parameters has a description of the glacier melt processes. The authors have somehow combined the snow and glacier melt and I am afraid that the true glacier melt is not represented as shown in (Wijngaard et al. 2017, Shrestha and Nepal, 2019, Lutz et al. 2014)
- What is the basis for lapse rate approach of this research? It would be better if you provide reference of the approach.
- Line 38, section 3.4: NSE has not been stated before. Are NSE and NS different? There should be consistency.
- Table 2: The efficiency criteria 'KGE' has not been mentioned before. KGE must be described in section 3.4.
- Result and Discussion: Discussion part is missing.

- The change in precipitation under RCP8.5 from mid to late century is from 6% to 4.6% in abstract. However, in conclusion, the reverse has been written. It should be corrected before publishing.
- Line 32, Page 3 of 16: What 'middle flows' represents? Interflow?
- Page 14: line: 45-47: for which period (mid century or end of the century)
- Page 14, line 48-51: what is the reason of decrease in flow in the late century than mid century.
-

Minor comments

- In Figure 1, Figure 2, and Figure 3, in scale bar, 'Km' should be written as 'km'.
- In Figure 4 and Figure 5, use of same line width for observed and simulated flow would make the hydrograph more comparable.
- Extensive correction of English is required before publishing. Some of the corrections (but not limited to) required are:
 1. Abstract, Line 42: R^2 and NSE equals 0.85 **each** for daily flow.
 2. Page 2 of 16
 - Line 56: More than 50% of world water demand **is** fulfilling by the rivers.
 - Line 58: What are the sources of the statement: 'Moreover, the climate system of the earth had changed up to large extent in the past' ?
 3. Page 3 of 16
 - Line 1: On average, **an increase of 0.74° C on Earth's** surface temperature was observed.
 - Line 9: fed by the river Indus and **it's** tributaries. Hydro-elctric or hydroelectricity?
 - Line 12: Reservoir **lives** are concerned
 - Line 15: had occurred; swift **of** population growth
 - Line 16: Removing hyphen between 'climate-change' would be better.
 - Line 22: 'Change in' before water temperature.
 - Line 31: simulate stream flow for **early**, mid, and late century.
 - Line 33: 's' in shift NOT 'S'
 4. Page 4 of 16
 - Line 11: Indus can be classified NOT classifies
 - Line 13: UIB extends up to 1150 km.
 - Line 14: perennial glacial ice-**cover area**
 - Line 18: 90% **of** catchment area
 - Line 22: Replace '&' by 'and'
 - Line 29 and 30: There is repition which has already mentioned in last paragraph of introduction.
 5. Page 7 of 16
 - Line 60: successfully **overlaying**.
 6. Page 14 of 16
 - Line 34: Replace 'affect' by 'effect'.

Additional references which could be cited in the paper

- Chevallier, P., Arnaud, Y., & Ahmad, B. (2011). Snow cover dynamics and hydrological regime of the Hunza River basin, Karakoram Range, Northern Pakistan.
- Shrestha, S., & Nepal, S. (2019). Water Balance Assessment under Different Glacier Coverage Scenarios in the Hunza Basin. *Water*, 11(6), 1124.

- Shrestha, M.; Koike, T.; Hirabayashi, Y.; Xue, Y.; Wang, L.; Rasul, G.; Ahmad, B. Integrated simulation of snow and glacier melt in water and energy balance-based, distributed hydrological modeling framework at Hunza River Basin of Pakistan Karakoram region. *J. Geophys. Res. Atmos.* 2015, 120, 4889–4919.
- Wijngaard, R. R., Lutz, A. F., Nepal, S., Khanal, S., Pradhananga, S., Shrestha, A. B., & Immerzeel, W. W. Future changes in hydro-climatic extremes in the Upper Indus, Ganges, and Brahmaputra River basins. *PLoS one*, 2017. 12(12): p. e0190224.

Appendix C

Predicting hydrologic responses to climate changes in highly glacierized and mountainous region Upper Indus Basin

The paper looked at the impact of climate change on river flows in the Upper Indus basin using SWAT hydrological model which was forced with high resolution regional climate model (RCMs) for two future scenarios RCP4.5 and 8.5. The model has able to replicate the historic reference period well, based on which the future hydrology is quantified. Under different scenarios, the future flows are likely to increase.

Response:

The authors would like to appreciate the reviewers' time and effort for providing us her/his valuable comments.

Major comments

Comment 1:

The authors fail to cite important publications on the hydrological aspect of the Upper Indus Basin. I have listed a few important here (but I encourage authors to look more)

- Shrestha, S., Nepal, S (2019)....
- Shrestha Maheshwar (2015).....

Response:

Yes, all of the mentioned and some other important papers have been cited in introduction and discussion section (please see in Pages 2 and 14).

Comment 2:

I think it would be very useful if authors compare the results with these published results to see where the historical hydrological modelling from this study compares

Response:

Yes, the above information has been added and comparison is drawn in discussion section (please see Page 14).

Comment 3:

It is not clear how the glacier response in the future in the model? Is glacier area changes due to the response of temperature change in the future (e.g. Wijngaard et al. 2017), Shrestha and Nepal, 2019 also showed the different response to streamflow under various glacier recession scenarios.

Response:

To incorporate the contribution of snow/glacier melt in the model, we used Temperature Index approach [1] with elevation bands. This approach allows the model to replicate the response of snow/glacier in a glaciated watershed by distributing the sub basins to several elevation bands. SWAT model provides melt water at each sub basin including the contribution of melt water from both snow and glaciers [2]. SFTMP,

SMTMP, SMFMN, SMFMX, TIMP, SNO50COV and SNOCOVMX are some parameters used by SWAT model for melt contribution from snow/glaciers. In SWAT model, the glacier melt can be accounted using snowmelt parameters such as depletion curves and temperature lag factor. All these parameters were successfully incorporated into the model and the information about the parameters were collected from literature study. Furthermore, detailed information can be found in Neitsch et al., [3].

The above information has been added (please see Section 3.1, Page 7).

Comment 4:

The methodology of using one RCM from CORDEX is not strong methodology, as the bias within one model could be high. There are many studies which look at the ensemble approach of few models.

Response:

There are many research studies that considered several GCMs or RCMs. But in a study, conducted by Khan and Koch, 2018 [4], a step-wise methodology was adopted for shortlisting and selection of suitable climate models for the Upper Indus Basin, based on a range of projected mean and extreme changes and skill in reproducing the past climate. The study was concentrated on RCA4 regional climate models from CORDEX. Out of all the selected CORDEX-South Asia experiments, the RCM “NorESM1-M_RCA4” was the most appropriate as depiction of the median tendencies of future climate for UIB. Hence, the NorESM1-M_RCA4 was selected for the present study.

Description of the RCM used in this study

S.No	Experiment		Driving AOGCM	RCM	RCM description
	Name	Short form			
1	NorESM1-M_RCA4	NOR	Nor-ESM1-M	RCA4	Rosby Centre regional atmospheric model version 4 (RCA4) [5]

The above information has been added (please see Section 2.2.4, Page 6).

Comment 5:

The increase in future flow is attributed to what? Increase in rainfall or increase in accelerated melting from glacier. It is not clear how the future temperature has affected glacier response. Is the SWAT glacier incorporation (if yes) is dynamic (changes in glacier area in the future) and stagnant?

Response:

Based on our results, a projected change in temperature varies between 2.36°C to 3.5°C under RCP4.5 (from mid-century to late century) and 2.92°C to 5.23°C under RCP8.5 (from mid-century to late century) was observed. Similarly, the expected changes in precipitation ranges from 2.4% to 2.5% for RCP4.5 and 6% to 4.6% under RCP8.5 (from mid-century to late century). It is clear that the increase in the future flows may be attributed to the increased temperature and also the precipitation extremes.

In the Upper Indus Basin, the increased temperature contributed to increased melt rates which rises the streamflow [6]. The same results were reported by Garee, K. et al., [7] that annual flow will increase ranging from 7.57% to 32.12% till the end of 21st century with a temperature increase from 1.39°C to 6.58°C. In our study, the flow increases in both mid (2041-2070) as well as in late century (2071-2100). The increased flow is due to high melt rates and the algorithm of snow/glacier melt was successfully incorporated in the SWAT model in the form of temperature index [1] with elevation bands. The SWAT model allows the sub basins to further split into ten elevation bands and the snow/glacier melt processes are simulated individually. The snow melt parameters (SMFMX, SMFMN), temperature lag factor (TIMP), snow fall temperature (SFTMP) and snow melt temperature (SMTMP) are allowed to vary spatially within the sub basins.

The above information has been added (please see Section 5, i.e., discussion, Page 14).

Comment 6:

It is not clear how the authors have implemented glacier melt in the SWAT model. Authors have used the snow/glacier melt but it is not clear whether snow and glacier melt are deal in a different way, as both have different melt processes; snow is more seasonal and based on available snow storage, but glacier remains available even the snow is gone.

Response:

Yes the snow and glacier are different melt processes. In the current study, the temperature index approach [1] with elevation band is used to simulate the snow/glacier melt and its contribution to streamflow. SWAT model provides melt water at each sub basin including the contribution from both snow and glaciers. The glacier melt in SWAT can be accounted using snowmelt parameters such as depletion curves and temperature lag factor. All these parameters were used in the model.

Similarly, the temperature index approach has been used by babur et al., [8] in Jhelum river basin, Garee et al., [7] in Hunza river basin, Anjum et al., [9] in swat river basin and zhang et al., [10] in yellow river basin for streamflow simulation in glaciated watershed.

The above information has been added (please see Section 3.1, Page 7).

Comment 7:

Page 10, Line 35: Table 3: None of the parameters has a description of the glacier melt processes. The authors have somehow combined the snow and glacier melt and I am afraid that the true glacier melt is not represented as shown in (Wijngaard et al. 2017, Shrestha and Nepal, 2019, Lutz et al. 2014)

Response:

In our study, the snow and glacier melt processes are dealt with the help of temperature index combined [1] with elevation band approach.

The above mentioned authors used different models in their studies. In the present study, we used the Soil & Water Assessment Tool (SWAT) which is a river basin scale model developed to quantify the effect of land management practices in large and complex watersheds. In SWAT model, there are some parameters related to and most sensitive to glacier and snowmelt i.e. SFTMP, SMTMP, SMFMN, SMFMX,

TIMP, SNO50COV and SNOCOVMX. The SFTMP (snowfall temperature) is used as a threshold to categorize the precipitation as rainfall or snowfall. When the air temperature is lower than SFTMP, the precipitation is classified as snowfall [2]. The snowpack will not melt until the snowpack temperature exceeds the threshold value i.e. SMTMP (snowmelt base temperature). SMFMX is the melt factor for 21 June and SMFMN is the melt factor for 21 December [2]. SNOCOVMX represents the snow water content agreeing with watershed full snow cover [7, 11]. The SNO50COV parameter which is volume of snow water that relates to 50% snow cover, can be used to adjust shape of depletion curve [12]. The TIMP is an empirical parameter used for snow density, extent, depth and other factors affecting snow pack temperature [11]. TIMP is used to regulate temperature effect on previous days based on current temperature of snow pack.

The above information has been added (please see Section 4.1, Pages 10 and 11).

Comment 8:

What is the basis for lapse rate approach of this research? It would be better if you provide reference of the approach.

Response:

One of the co-author, Asim Jahangir Khan [13] used a new method for interpolation and correction of the data (lapse rate approach) across the whole Upper Indus Basin considering the orographic effect in the region and specifically correcting the influences induced by higher elevation, glacier mass-balance and actual evapotranspiration. In the present study, we used the same lapse rate approach.

The above information has been added (please see Section 3.1, Page 7).

Comment 9:

Line 38, section 3.4: NSE has not been stated before. Are NSE and NS different? There should be consistency.

Response:

Both NS and NSE are same. Now NS is replaced with NSE and is consistent throughout the manuscript.

The above information has been added (please see Section 3.4, Page 8)

Comment 10:

Table 2: The efficiency criteria 'KGE' has not been mentioned before. KGE must be described in section 3.4.

Response:

The Kling-Gupta Efficiency (KGE) is a parameter based on NSE and mean squared error. It shows the correlation, relative variability and bias between observed and simulated values [14]. Alike NSE, the KGE ranges from $-\infty$ to 1 with the optimal value as 1 [15].

The above information has been added (please see Section 3.4, Page 8).

Comment 11:

Result and Discussion: Discussion part is missing.

Response:

The discussion part has been added separately (please see Section 5, Page 14).

Comment 12:

The change in precipitation under RCP8.5 from mid to late century is from 6% to 4.6% in abstract. However, in conclusion, the reverse has been written. It should be corrected before publishing.

Response:

The above information has been corrected (please see conclusion Section, Page 15).

Comment 13:

Line 32, Page 3 of 16: What 'middle flows' represents? Interflow?

Response:

The term "middle flows" was used by researchers Babur, M., et al., [8]. The Information has been corrected (please see section 1, Introduction, Page 2).

Comment 14:

Page 14: line: 45-47: for which period (mid-century or end of the century)

Response:

A warmer climate is expected for the study area in future with a projected change in temperature between 2.36°C to 3.5°C under RCP4.5 (from mid-century to late century) and 2.92°C to 5.23°C under RCP8.5 (from mid-century to late century).

Likewise, the anticipated changes in precipitation ranges from 2.4% to 2.5% for RCP4.5 and 6% to 4.6% under RCP8.5 (from mid-century to late century).

The above information has been added in conclusion section (please see Page 15).

Comment 15:

Page 14, line 48-51: what is the reason of decrease in flow in the late century than mid-century?

Response:

This is due to the fact that, in the late-century, the increase contribution to the flows by snow-melt (due to higher temperatures) is probably balance out and offset by the reduction of snowfall and thus accumulation [16] and the increased rates of evapotranspiration, which act as a further loss from the water balance and result in reduced annual flows under RCP8.5 in late century [16] [17].

The above information has been added in conclusion section (please see Page 15).

Minor comments

- In Figure 1, Figure 2, and Figure 3, in scale bar, 'Km' should be written as 'km'.
- In Figure 4 and Figure 5, use of same line width for observed and simulated flow would make the hydrograph more comparable.
- Extensive correction of English is required before publishing. Some of the corrections (but not limited to) required are:

1. Abstract, Line 42: R^2 and NSE equals 0.85 each for daily flow.

2. Page 2 of 16

Line 56: More than 50% of world water demand is fulfilling by the rivers.

Line 58: What are the sources of the statement: 'Moreover, the climate system of the earth had changed up to large extent in the past'?

3. Page 3 of 16

Line 1: On average, an increase of 0.74°C on Earth's surface temperature was observed.

Line 9: fed by the river Indus and its tributaries. Hydro-electric or hydroelectricity?

Line 12: Reservoir lives are concerned

Line 15: had occurred; swift of population growth

Line 16: Removing hyphen between 'climate-change' would be better.

Line 22: 'Change in' before water temperature.

Line 31: simulate stream flow for early, mid, and late century.

Line 33: 's' in shift NOT 'S'

4. Page 4 of 16

Line 11: Indus can be classified NOT classifies

Line 13: UIB extends up to 1150 km.

Line 14: perennial glacial ice-cover area

Line 18: 90% of catchment area

Line 22: Replace '&' by 'and'

Line 29 and 30: There is repetition which has already mentioned in last paragraph of introduction.

5. Page 7 of 16

Line 60: successfully overlaying.

6. Page 14 of 16

Line 34: Replace 'affect' by 'effect'.

Response:

All the suggestions have been incorporated.

Additional references which could be cited in the paper

- Chevallier, P., Arnaud, Y., & Ahmad, B. (2011). Snow cover dynamics and hydrological regime of the Hunza River basin, Karakoram Range, Northern Pakistan.
- Shrestha, S., & Nepal, S. (2019). Water Balance Assessment under Different Glacier Coverage Scenarios in the Hunza Basin. *Water*, 11(6), 1124.

- Shrestha, M.; Koike, T.; Hirabayashi, Y.; Xue, Y.; Wang, L.; Rasul, G.; Ahmad, B. Integrated simulation of snow and glacier melt in water and energy balance-based, distributed hydrological modeling framework at Hunza River Basin of Pakistan Karakoram region. *J. Geophys. Res. Atmos.* 2015, 120, 4889–4919.
- Wijngaard, R. R., Lutz, A. F., Nepal, S., Khanal, S., Pradhananga, S., Shrestha, A. B., & Immerzeel, W. W. Future changes in hydro-climatic extremes in the Upper Indus, Ganges, and Brahmaputra River basins. *PIOS one*, 2017. 12(12): p. e0190224.

Response:

All the suggested papers have been cited.

REFERENCES (cited in this document)

1. Hock, R., *Temperature index melt modelling in mountain areas*. *Journal of hydrology*, **2003**. 282(1-4): p. 104-115.
2. Neitsch, S. L., Arnold, J. G., Kiniry, J. R., & Williams, J. R. **2011**. Soil and water assessment tool theoretical documentation version 2009. Texas Water Resources Institute.
3. Neitsch, S., et al., **2005**. *Soil and water assessment tool theoretical documentation version 2005. Grassland*. Soil and Water Research Laboratory, Blackland Research Center, Temple, Texas.
4. Khan, A. and M. Koch, *Selecting and Downscaling a Set of Climate Models for Projecting Climatic Change for Impact Assessment in the Upper Indus Basin (UIB)*. *Climate*, **2018**. 6(4): p. 89.
5. Samuelsson, P., Jones, C. G., Will' En, U., Ullerstig, A., Gollvik, S., Hansson, U. L. F., & Wyser, K. (**2011**). The Rossby Centre Regional Climate model RCA3: model description and performance. *Tellus A: Dynamic Meteorology and Oceanography*, 63(1), 4-23.
6. Lutz, A. F., Immerzeel, W. W., Shrestha, A. B., & Bierkens, M. F. P. **2014**. Consistent increase in High Asia's runoff due to increasing glacier melt and precipitation. *Nature Climate Change*, 4(7), 587-592.
7. Garee, K., Chen, X., Bao, A., Wang, Y., & Meng, F. **2017**. Hydrological modeling of the upper indus basin: A case study from a high-altitude glacierized catchment Hunza. *Water*, 9(1), 17.
8. Babur, M., Babel, M. S., Shrestha, S., Kawasaki, A., & Tripathi, N. K. **2016**. Assessment of climate change impact on reservoir inflows using multi climate-models under RCPs—The case of Mangla Dam in Pakistan. *Water*, 8(9), 389.
9. Anjum, M.N., Y. Ding, and D. Shangguan., **2019**. *Simulation of the projected climate change impacts on the river flow regimes under CMIP5 RCP scenarios in the westerlies dominated belt, northern Pakistan*. *Atmospheric Research*,. 227: p. 233-248.
10. Zhang, Y., Su, F., Hao, Z., Xu, C., Yu, Z., Wang, L., & Tong, K. **2015**. Impact of projected climate change on the hydrology in the headwaters of the Yellow River basin. *Hydrological Processes*, 29(20), 4379-4397.
11. Lemonds, P.J. and J.E. McCray., **2007**. *Modeling Hydrology in a Small Rocky Mountain Watershed Serving Large Urban Populations 1*. *JAWRA Journal of the American Water Resources Association*,. 43(4): p. 875-887.
12. Noor, H., Vafakhah, M., Taheriyoun, M., & Moghadasi, M. **2014**. Hydrology modelling in Taleghan mountainous watershed using SWAT. *Journal of Water and Land Development*, 20(1), 11-18.
13. Khan, A. and M. Koch., **2018**. *Correction and informed regionalization of precipitation data in a high mountainous region (Upper Indus Basin) and its effect on SWAT-modelled discharge*. *Water*,. 10(11): p. 1557.

14. Gupta, H. V., Kling, H., Yilmaz, K. K., & Martinez, G. F. **2009**. Decomposition of the mean squared error and NSE performance criteria: Implications for improving hydrological modelling. *Journal of hydrology*, 377(1-2), 80-91.
15. Franco, A.C.L. and N.B. Bonumá,. **2017**. *Multi-variable SWAT model calibration with remotely sensed evapotranspiration and observed flow*. RBRH,. 22.
16. Khan, A.J., **2018**. *Estimating the Effects of Climate Change on the Water Resources in the Upper Indus Basin (UIB)*., Universitätsbibliothek Kassel.
17. Peters, G. P., Andrew, R. M., Boden, T., Canadell, J. G., Ciais, P., Le Quéré, C., & Wilson, C. **2012**. The challenge to keep global warming below 2 C. *Nature Climate Change*, 3(1), 4.

Appendix D

Review of Predicting Hydrologic Responses to Climate Change in Highly Glacierized and Mountainous Region Upper Indus Basin by Shah et al.

This manuscript documents an application of climate downscaling to upper Indus basin. The problem setup is generally Ok, but writing needs to be significant improved. Currently there are many typos and grammar errors. I only give several examples below

P3, L15, "...severe storms and floods have been occurred in its history" should be "severe storms and floods have occurred in its history"

P3, L16-17, "Besides, the swift population growth and the accompanying land-use alterations further intensifying the problems induced by climate-change ..." should be "Besides, the rapid population growth and the accompanying land-use alterations further intensify the problems induced by climate-change"

P4, L10-L11, "The River Indus can be classifies as one of the major river in Asia,..." => "The Indus River can be classified as one of the major rivers in Asia,"

All the suggested changes have been incorporated please.

Appendix E

Respected Editor and Reviewers,

We, the authors, have revised the manuscript in the light of your comments. We would like to thank you for your valuable suggestions and comments. The responses to reviewer's comments are as follows. Please note that our response (addition/modification) is in blue colored text in the manuscript.

Regards,
The Authors

Comment 1.

The introduction and discussion now provide greater context and links with existing literature. The comparison with other work (P14) is especially informative. As a minor addition, I think it would be useful to include a sentence towards the end of section 1, indicating clearly how this study builds upon those existing studies cited. Essentially, a clear statement of novelty would be useful here.

Response:

Thanks a lot for the suggestions to improve section 1. The following text has been added to the paper to indicate the novelty and innovation presented by the current work.

The calculations of water balance and spatially distributed rainfall-runoff models require high resolution climatic datasets i.e. temperature and precipitation. The UIB is facing the same problem because of valley based gauging stations which causes temporal discontinuities and unable to capture orographic influences. Therefore, the climate studies in such mountainous regions are facing the water imbalances [1].

The present study utilized the fully distributed, regionalized and corrected precipitation and temperature data [1] which was interpolated to sub-basin centroids and constructed based on true situation in UIB. The datasets were corrected by incorporating the orographic effect and improving the influences induced by higher elevation, available runoff data, glacier mass-balance and actual evapotranspiration. The precipitation data is the most important input in modelling studies of mountainous regions [2, 3] and the results are strongly affected because of uncertainty in spatial distribution. [1]. Furthermore, in this study the elevation bands were used for better representation of the melt processes and assigned distributed melt parameters to different bands and sub-basins.

The above information has been added at the end of introduction section please. Kindly see Section 1, Page 3.

Comment 2.

I think there remains a need for greater clarity around the representation of glaciers in the model. The reviewer rightly highlighted this as a point requiring further description, given the expected significance of glacier melt in the future hydrology of the study basin. The reviewer questioned a number of times whether or not the glaciers are represented dynamically in the model, and I remain unsure of the answer even having read through the new corrections. I think this point needs to be addressed prior to publication.

Response:

Although SWAT hydrological model can handle and simulate rainfall–runoff processes and snow hydrology, the glacio-hydrological processes are not include as part of the official version [4]. The contribution of snow/glacier melt in the model was incorporated using the Temperature Index approach [5] with elevation bands. This approach allows the model to replicate the response of snow/glacier in a glaciated watershed by distributing the sub basins to several elevation bands. Although, the SWAT model assume the glaciers area as static [6], it provides melt water at each sub basin including the contribution of melt water from both snow and glaciers [7] based on Temperature Index approach with elevation bands and the parameters such as SFTMP, SMTMP, SMFMN, SMFMX, TIMP, SNO50COV and SNOCVMX etc. All these parameters were successfully incorporated into the model and the information about the parameters were collected from literature study. Furthermore, detailed information can be found in Neitsch, S., et al., [8].

The authors of the present research are now focusing for incorporation of a dedicated glacier module or coupling of a glacier model with SWAT for study of glaciers and its response in the catchment.

The above details and information on Snow module has been added please. Kindly see Section 3.1 (Description of the SWAT model) Page 7 and conclusion section page 16.

References (Cited in this document)

1. Khan, A. Koch, M. Correction and informed regionalization of precipitation data in a high mountainous region (Upper Indus Basin) and its effect on SWAT-modelled discharge. *Water*, **2018**. 10, 1557.
2. Shrestha, S. Nepal, S. Water Balance Assessment under Different Glacier Coverage Scenarios in the Hunza Basin. *Water*, **2019**. 11, 1124.
3. Garen, D.C. Marks, D. Spatially distributed energy balance snowmelt modelling in a mountainous river basin: estimation of meteorological inputs and verification of model results. *Journal of Hydrology*, **2005**. 315, 126-153.

4. Wang, X., Luo, Y., Sun, L., Zhang, Y. Assessing the effects of precipitation and temperature changes on hydrological processes in a glacier-dominated catchment. *Hydrological Processes*, **2015**. 29, 4830-4845.
5. Hock, R., Temperature index melt modelling in mountain areas. *Journal of hydrology*, **2003**. 282, 104-115.
6. Yin, Z., Feng, Q., Liu, S., Zou, S., Li, J., Yang, L., Deo, R. C. The spatial and temporal contribution of glacier runoff to watershed discharge in the Yarkant River Basin, Northwest China. *Water*, **2017**. 9, 159.
7. Neitsch, S. L., Arnold, J. G., Kiniry, J. R., Williams, J. R. Soil and water assessment tool theoretical documentation version 2009. **2011**, *Texas Water Resources Institute*.
8. Neitsch, S. L., Arnold, J. G., Kiniry, J. R., Williams, J. R. Soil and water assessment tool theoretical documentation version 2005. Grassland. Soil and Water Research Laboratory, Blackland Research Center, Temple, Texas, **2005**.